# Artificial dual solid-electrolyte interfaces based on in situ organothiol transformation in lithium sulfur battery

Wei Guo[1,4], Wanying Zhang[1,4], Yubing Si[1], Donghai Wang [2], Yongzhu Fu [1✉] & Arumugam Manthiram[3]

The interfacial instability of the lithium-metal anode and shuttling of lithium polysulfides in lithium-sulfur (Li-S) batteries hinder the commercial application. Herein, we report a bifunctional electrolyte additive, i.e., 1,3,5-benzenetrithiol (BTT), which is used to construct solid-electrolyte interfaces (SEIs) on both electrodes from in situ organothiol transformation. BTT reacts with lithium metal to form lithium 1,3,5-benzenetrithiolate depositing on the anode surface, enabling reversible lithium deposition/stripping. BTT also reacts with sulfur to form an oligomer/polymer SEI covering the cathode surface, reducing the dissolution and shuttling of lithium polysulfides. The Li–S cell with BTT delivers a specific discharge capacity of 1,239 mAh g$^{-1}$ (based on sulfur), and high cycling stability of over 300 cycles at 1C rate. A Li–S pouch cell with BTT is also evaluated to prove the concept. This study constructs an ingenious interface reaction based on bond chemistry, aiming to solve the inherent problems of Li–S batteries.

[1] College of Chemistry, Zhengzhou University, Zhengzhou, PR China. [2] Department of Mechanical Engineering, The Pennsylvania State University, University Park, PA, USA. [3] Materials Science and Engineering Program & Texas Materials Institute, The University of Texas at Austin, Austin, TX, USA. [4]These authors contributed equally: Wei Guo, Wanying Zhang. ✉email: yfu@zzu.edu.cn

The lithium-metal anode is recognized as the "Holy Grail" for rechargeable batteries on account of its high theoretical specific capacity (~3860 mAh g$^{-1}$) and low redox potential (−3.04 V vs. NHE). Elemental sulfur is a promising cathode material with a similarly large theoretical capacity of 1,675 mAh g$^{-1}$. Lithium-sulfur (Li–S) batteries couple a Li anode with a sulfur (S) cathode, enabling high specific energy (~2600 Wh kg$^{-1}$) and great potential to be used for energy-intensive applications. However, it is plagued with two key barriers that limit its applications: interfacial instability of lithium-metal anode and shuttling of soluble intermediate poly-sulfides of the sulfur cathode[1]. To tackle these problems, several techniques have been proposed, including: electrode modification, such as designing structured Li-metal anodes[2–4]; confining S within porous carbon or other nano-architectures with tailored surface[5–7]; electrolyte modification, such as using redox mediators and novel lithium salt/ionic liquid electrolyte[8–14]; electrode-electrolyte interface modification[15], such as constructing solid-electrolyte interphase (SEI) on the Li-metal surface[16–23]; and protective layers on the S cathode[24,25]. These efforts are directly related to the electrode-electrolyte interfaces[26–31], which have profound effect on battery performance. However, most endeavors have so far only focused on a single electrode, not both. To fully optimize the anode and cathode requirements, electrolyte additives capable of forming interfaces with both anode and cathode are the most promising solutions[32,33]. Suitable electrolyte additives should be able to undergo electrochemical transformations on both electrodes and form functional surface films to solve the both electrodes requirements.

Organic materials have attracted significant attention due to the relative abundance of the constituent elements and highly tunable functional groups compared to inorganic compounds[34,35]. Organic compounds have been widely studied to establish electroactive structures, understand the charge/ion transport mechanism, and explore interfacial electrochemical transformations[36,37]. Recent advances reveal their tremendous potential in Li–S batteries. The introduction of organic compounds containing functional groups into sulfur (selenium) electrodes could lead to interesting electrochemical transformations, trigger molecular recombination, change the path of electrochemical redox in battery, and ultimately lead to improved electrochemical performance beyond the intrinsic disadvantages of traditional materials[38]. In particular, organosulfide materials have shown unlimited potential as active materials alone or as electrolyte additives in Li–S batteries. Particularly, organothiols consisting of organic molecules with SH-groups are well suited to fabricate structurally layers. Organothiols are typically used as precursors for the synthesis of organopolysulfides, can react with lithium metal and elemental sulfur, thus they are also promising electrolyte additives. 1,3,5-benzenetrithiol (BTT), as a typical organothiol, has the symmetrical sulfhydryl groups on the benzene ring, which can not only react with Li to form SEI on the Li metal anode but also self-assemble with sulfur on the sulfur cathode to form stable and adaptable monolayer over homogenous areas.

Here, we report the promising electrolyte additive, i.e., BTT, to fabricate dual SEIs (D-SEIs), which involves an exchange reaction on the Li-metal anode and electrochemical polymerization on the sulfur cathode. Our approach was realized by in situ reactions of BTT via the highly reactive S–H groups with Li and S to produce interfacial layers comprised of S–Li and S–S bonds. The in situ formed S-X (where X is Li or S) bonds regulate the Li deposition/stripping behavior and change the redox path of the sulfur cathode, leading to enhanced performance. The Li–S cell with BTT delivers a discharge capacity of 1,239 mAh g$^{-1}$ (based on sulfur) and high cycling stability of over 300 cycles at 1C rate. The lithium symmetric cell with the BTT maintains a low deposition potential at about 40 mV over 270 h with the current density of 1

mA cm$^{-2}$ and capacity density of 1 mAh cm$^{-2}$. BTT is also successfully applied in a Li–S pouch cell. The in situ S–X (S or Li) bond formation and interfacial electrochemical transformation mechanism are investigated systematically.

## Results

**Formation of D-SEIs.** The formation of D-SEIs starts with in situ chemical transformation of the S–H groups of BTT. As we know, thiols are potential hydrogen-atom donors and functional chain carriers. The abstraction of free-radical hydrogen from S–H bond is accounted for by this fact. Thus, they are used as trapping agents for biradicals. In addition, the weak bonding between sulfur and hydrogen makes thiols fairly acidic (pKa of about 11 for the thiol). The sulfhydryl group is acidic enough to react with alkalis to form thiolate salts, which has been confirmed by previous work[39]. Thiols can be oxidized to disulfides through treatment with a mild oxidant, such as sulfur. Herein, BTT is chosen as the additive in the ether electrolyte due to its excellent chemical reactivity with Li and sulfur.

BTT from ether electrolyte immediately undergoes reactions with the lithium/sulfur, forming new compounds, as shown in equations (1) and (2).

Scaling-up two reactions separately, the kinetics of reactions are very fast, which are proved by experimental data and preliminary works[40,41]. When testing the reaction between BTT and the cathode material sulfur, H$_2$S was detected quickly by the lead acetate test paper upon mixing BTT and sulfur (Supplementary Fig. 1a, b) (magnified about 90 times). Meanwhile the color of the solution with BTT changed within a few seconds upon putting a piece of lithium metal into the solution, which is accompanied by small gas bubbles. (Supplementary Fig. 1c) (magnified about 95 times). The product was confirmed to be lithium benzentrithio-late (Li$_3$-BTT) by $^7$Li nuclear magnetic resonance ($^7$Li-NMR) (Supplementary Fig. 1d). Both reactions (1) and (2) generate gases, which greatly promote the completion of the reactions. Based on this, reactions (1) and (2) were performed in situ in coin-type cells utilizing lithium foil and sulfur as the negative and positive electrodes, respectively. The properties of D-SEIs, alongside the electrochemical formation and electrochemical transformation mechanisms, were investigated systematically.

**Electrochemical evaluation of Li-S cells with D-SEIs.** Li-S cells were fabricated using 1 M LiTFSI in DOL/DME (1:1 v/v) with 0.15 M LiNO$_3$ as the electrolyte. Cells were made with and without BTT as the bifunctional additive, with the cell without BTT serving as the control. As shown in Supplementary Fig. 2a, white powder BTT can dissolve in Li–S electrolyte instantly and form the BTT electrolyte. Cycling performance and the charge and discharge curves for Li-S with different concentration additive at 0.5C rate are shown in Supplementary Fig. 2b–d, Li–S with 0.15 M BTT shows the lowest overpotential and best capacity retention, then the 0.15 M BTT electrolyte was used in the work. The typical charge/discharge profile of the Li–S cells with and without BTT at 0.2C rate are shown in Fig. 1a. The Li–S cell with BTT (designated the BTT cell) exhibits three discharge voltage

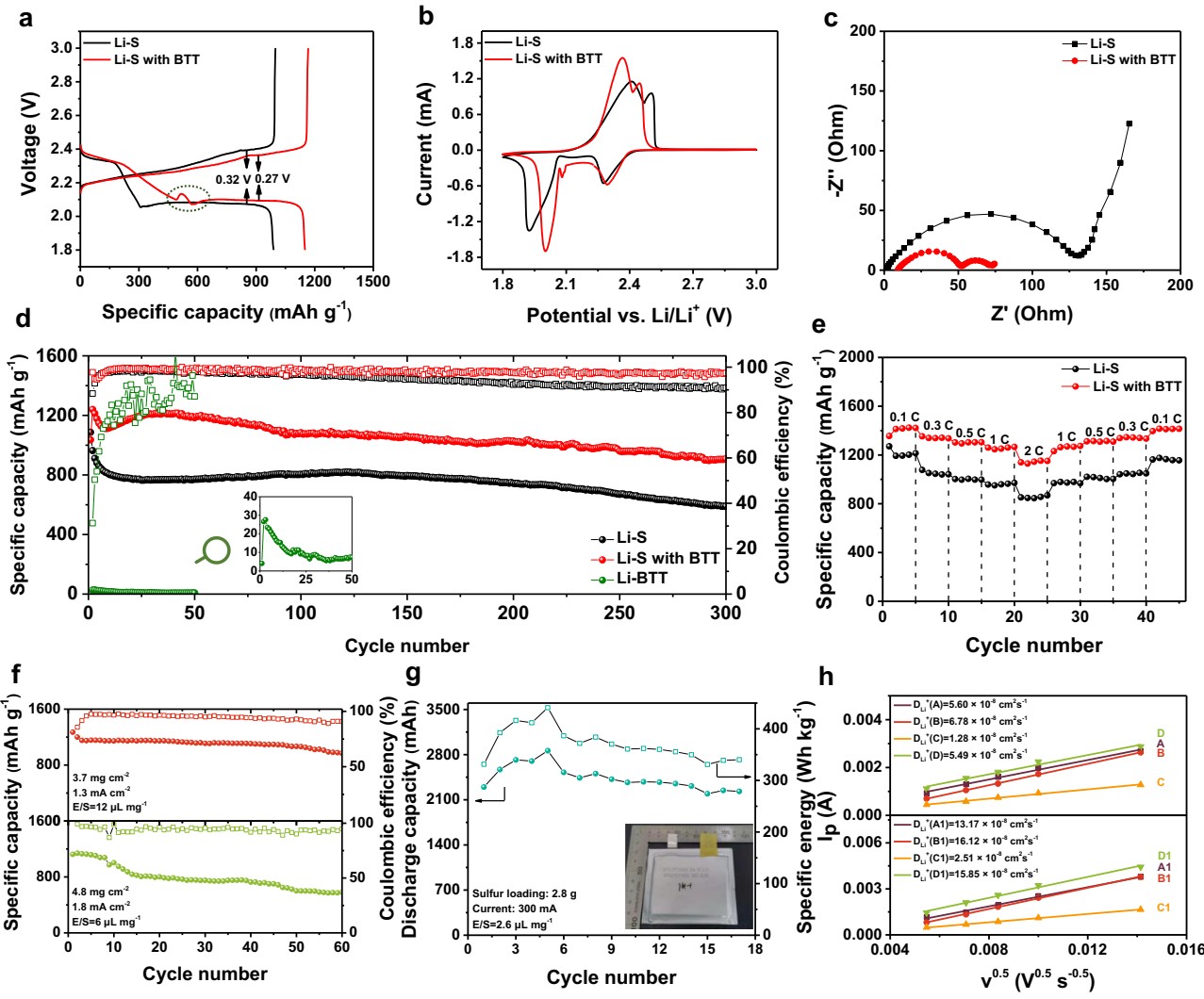

**Fig. 1 Electrochemical performance of Li-S cells with BTT and blank electrolyte. a** The 5th (stable cycle) galvanostatic charge/discharge profile of Li-S cells with and without BTT at 0.2C rate. **b** Cyclic voltammograms (CV) scans of the two types of cells at 0.05 mV s⁻¹. **c** Nyquist plots of the two types of cells before cycling. **d** Long-cycling performance and Coulombic efficiency of the Li-S cells with and without BTT at 1C rate for 300 cycles. **e** The different C-rate performance of two types of cells at 0.1C, 0.3C, 0.5C, 1C, and 2C rates and then back to the low C rate. **f** Cycling performance of two BTT cells at 0.2C rate with high sulfur loadings of 3.7 and 4.8 mg cm⁻² and E/S ratios of 12 and 6 μL mg⁻¹, respectively. **g** Cycling performance of a BTT pouch cell with total sulfur mass of 2.8 g and E/S ratio of 2.6 μL mg⁻¹. The current is 300 mA. The specific energies are based on the masses of electrodes (lithium metal anode and sulfur cathode) and electrolyte. The inset photograph shows the front image and dimensions of the pouch cell. **h** Linear fits of the peak currents and the Li⁺ diffusion coefficient calculated by the Randles-Sevcik equation for Li-S cells with and without BTT.

plateaus including a plateau at 2.4 V, a steep slope in 2.3–2.1 V, a raised small peak at 2.1 V, and a long plateau at 2.1 V. The difference between BTT cell and the control cell (Li–S cell) is in the steep slope area. The BTT cell has a smaller slope, a larger discharge capacity, and a small bulge at 2.1 V. As we know, the traditional discharge curve of a Li–S cell is classified as: $S_8$ to $Li_2S_x$ (4≤x≤8) (from 2.4 V), $Li_2S_x$ (4≤x≤8) to $Li_2S_x$ (2≤x<4) (from 2.30 to 2.1 V), and $Li_2S$ is formed (2.1 V). In 2.30–2.1 V region of the BTT cell, in addition to the transition of $Li_2S_x$ (4≤x≤8), the S–S bonds of the BTT oligomer formed in the recharged process break, and bind with lithium ions and electrons, turning into $Li_3$-BTT and $Li_2S$. The formed $Li_3$-BTT completely precipitates at 2.1 V, as shown in Fig. 1a, which is overlapped with the traditional sulfur transition of $Li_2S_x$ (4≤x≤8) to $Li_2S_x$ (2≤x<4), causing the small peak at 2.1 V. On the contrary, because some of the sulfur is converted to $Li_2S$ in advance, the 2.1 V plateau is shorter than that of the Li–S cell. The voltage hysteresis of the BTT cell is

about 0.27 V, which is lower than that of the control cell (0.32 V). Cyclic voltammetry (CV) measurements confirm the variation observed in the charge and discharge profile (Fig. 1b). It is worth noting that the BTT cell has higher response current than the control cell. There is a small reduction peak contributed to $Li_3$-BTT at 2.1 V, which is consistent with the discharge curve. The electrochemical impedance spectroscopy (EIS) measurements shown in Fig. 1c illustrate the impedance for both control and BTT cells before cycling. The resistance of the BTT cell is much lower than that of the control cell. This suggests that BTT plays a key role in the formation of the surface films on the lithium metal anode and sulfur cathode. For the control cell, the impedance spectrum has a phase element, corresponding to the Li⁺ migration in the rough interfaces between the electrodes and electrolyte. For the BTT cell, it has two semicircles. The semicircle in the high frequency range is related to the Li⁺ migration through the D-SEIs formed by the chemical reactions on lithium and sulfur

cathodes. This indicates that the chemical reactions have already started when BTT were in contact with lithium and sulfur electrodes. With the infiltration of the electrolyte and cycles of the cell, a uniform interface is gradually formed in the BTT cell. The semicircle in the low frequency range is ascribed to the charge transfer across the D-SEIs. After 10 cycles and 50 cycles, the BTT cell still has lower impedance than that of the control cell (Supplementary Fig. 3a, b), mainly because of the fast transfer of $Li^+$ in the electrolyte owing to the $Li_3$-BTT and uniform SEIs owing to electrochemical reactions between BTT and lithium/sulfur. The detailed impedance of the cells is shown in Supplementary Table 1.

The cycling performance and Coulombic efficiency of the BTT cell at 1C rate are shown in Fig. 1d. The initial discharge capacity of the BTT cell is 1,036 mAh $g^{-1}$, which is lower than that in the second cycle (1,239 mAh $g^{-1}$) due to the participation of the additive. After 300 cycles, the BTT cell still has a high discharge capacity at 907 mAh $g^{-1}$ and the capacity retention is 87.6%. In contrast, the control cell has a lower initial discharge capacity of 1,086 mAh $g^{-1}$ and the capacity quickly drops to 770 mAh $g^{-1}$ after 20 cycles. It shows a continuous decrease in the discharge capacity which is only 589 mAh $g^{-1}$ after 300 cycles with a capacity retention of 54.3%. The Coulombic efficiency of the BTT cell is maintained at about 98% with little decrease after long cycles due to the stable D-SEIs. The Coulombic efficiency of the control cell is 90% after 300 cycles because of the constant consumption of $LiNO_3$. The charge/discharge curves of both cells and the galvanostatic charge/discharge voltage profile of the BTT cell at different cycles are shown in Supplementary Fig. 4a, b, respectively. The BTT cell has a lower overpotential and higher reversibility than the control cell, which yields a capacity retention of 84.7% after 200 cycles at 0.5C rate as shown in Supplementary Fig. 5. Compared to the recent work of additives in Li-S batteries (Supplementary Table 2 and Supplementary Fig. 6), the BTT cell has better capacity retention even after long cycles. In order to further measure the contribution of BTT to the capacity, BTT as the single electrochemically active cathode material was tested in the Li-BTT cell. The cycling performances at different C rates are shown in Supplementary Fig. 7a–c, which exhibit a rapid drop in capacity followed by a constant, low value after 20 cycles. Therefore, the BTT cell's high capacity and great reversibility is rooted in the D-SEIs originated from the S-H reaction.

To understand the practical performance of the BTT cell, the cyclability at different C rates for the two cells were studied (Fig. 1e). The BTT cell exhibits a stable capacity of 1420, 1330, 1300, 1250, and 1150 mAh $g^{-1}$ at 0.1C, 0.3C, 0.5C, 1C, and 2C rates, respectively, and maintains good recovery capability when the current returns to the low C rates. The performance obtained by the control cell fluctuates greatly with the C rate, and generally shows an attenuation trend. The improved performance of the BTT cell implies better electrode kinetics, which mainly results from the high $Li^+$ diffusion. Building high-energy-density Li-S batteries are also reflected by constructing high sulfur loading electrodes with a remarkably low electrolyte/sulfur ratio (E/S ratio). As depicted in Fig. 1f, electrodes with increased sulfur areal loadings of 3.7 and 4.8 mg $cm^{-2}$ (cycled at 0.2C rate, corresponding to the current density of 1.3 and 1.8 mA $m^{-2}$, respectively) deliver discharge capacities of 1200 and 1139 mAh $g^{-1}$ for the 2nd cycle, respectively. As cycling proceeds, the reversible discharge capacities maintain 970 and 581 mAh $g^{-1}$ after 60 cycles and capacity retentions are 76.4% and 51.7%, respectively. Corresponding charge/discharge voltage curves in the high loading BTT cells on the 5th cycle are shown in Supplementary Fig. 8a. The electrolyte/sulfur (E/S) ratios of these cells are 12 and 6 µL $mg^{-1}$, respectively. The obvious capacity decay for the BTT cell

with low E/S ratio is mainly due to the high porosity of the carbon paper as current collector. To further reduce the E/S ratio, the BTT electrolyte was applied in a Li-S pouch cell. The total sulfur mass in the cell is 2.8 g and the E/S ratio was reduced to 2.6 µL $mg^{-1}$. The pouch cell remains decent cycling performance for 17 cycles with the maximum capacity of 2861 mAh corresponding to the specific capacity of 1022 mAh $g^{-1}$ based on the mass of sulfur in the 5th cycle. The charge/discharge voltage profile is shown in Supplementary Fig. 8b. The maximum specific energy based on the masses of electrodes and electrolyte reaches 441 Wh $kg^{-1}$. After 17 cycles, the specific capacity is 796 mAh $g^{-1}$ and the specific energy is 340 Wh $kg^{-1}$. The pouch cell study proves that the BTT electrolyte can be successfully applied in large pouch cell which can work reasonably under the low E/S ratio.

To further verify the effect of the $Li^+$ diffusion in the cell, CV measurements were performed with different scan rates and the $Li^+$ diffusion coefficients were calculated by the Randles-Sevcik equation. The different CV measurements for the control cell are shown in Supplementary Fig. 9a. Four main peaks are assigned, with the anodic peaks labeled by A, B and the cathodic peaks labeled by C, D. Similarly, four peaks named A1, B1, C1, D1 belong to the BTT cell (Supplementary Fig. 9b). The corresponding linear fittings for the four peaks of the two cells are shown in Fig. 1h. The $Li^+$ diffusion coefficients for the four peaks of the BTT and control cells are calculated as shown in Supplementary Table 3. The $Li^+$ diffusion coefficients for the BTT cell are higher than those of the control cell. The increase of the $Li^+$ diffusion coefficient is related to two factors: the SEI formed on the anode and the dissolution of the discharge product of $Li_3$-BTT in the electrolyte. Hence, BTT enhances the dynamics of Li–S cells through maintaining smooth $Li^+$ diffusion and providing improved reaction kinetics.

**Properties of Li anode with BTT-original interfacial layers**. To evaluate the influence of the BTT interfacial layer on the electrochemical performance of Li-metal anode, symmetric Li cells were assembled using BTT electrolyte. In the case of the control cell, unmodified ether electrolyte was used. Figure 2a demonstrates the potential profile of the BTT symmetric cell and control cell at a constant current density of 0.5 mA $cm^{-2}$ with a capacity density of 0.5 mAh $cm^{-2}$. The BTT symmetric cell has a low and stable voltage hysteresis at about 12 mV over 400 h. However, a short circuit due to lithium dendrites occurred after the control cell was cycled steadily for 200 h. The enlarged profiles of the short circuit are shown in the right upper corner of Fig. 2a and Supplementary Fig. 10a. Increasing the current density to 1 mA $cm^{-2}$ with a capacity density of 1 mAh $cm^{-2}$ (Fig. 2b), the control symmetric cell expresses fluctuation and a high plating potential at about 150 mV, while the BTT symmetric cell has a stable deposition voltage at about 40 mV. The enlarged profiles of the plating/stripping curves at 1 mA $cm^{-2}$ (1 mAh $cm^{-2}$) are exhibited in the inset of Fig. 2b and Supplementary Fig. 10b. The BTT symmetric cell reveals a regular increase in voltage hysteresis with the increase of current density as shown in Fig. 2c. By contrast, the control symmetric cell exhibits an unstable voltage profile with fluctuation of voltage hysteresis. The corresponding partially enlarged detail of the voltage profiles at different current densities are displayed in Supplementary Fig. 10c–e. These data indicate that the interfacial layer formed on the lithium metal anode by the BTT additive provides stable and continuous protection and suppresses the production of Li dendrites under fast Li plating/stripping.

To verify the formation of SEI during the lithium plating/ stripping in symmetric Li cell, the lithium-metal electrodes of the two cells (BTT cell and control cell) were characterized

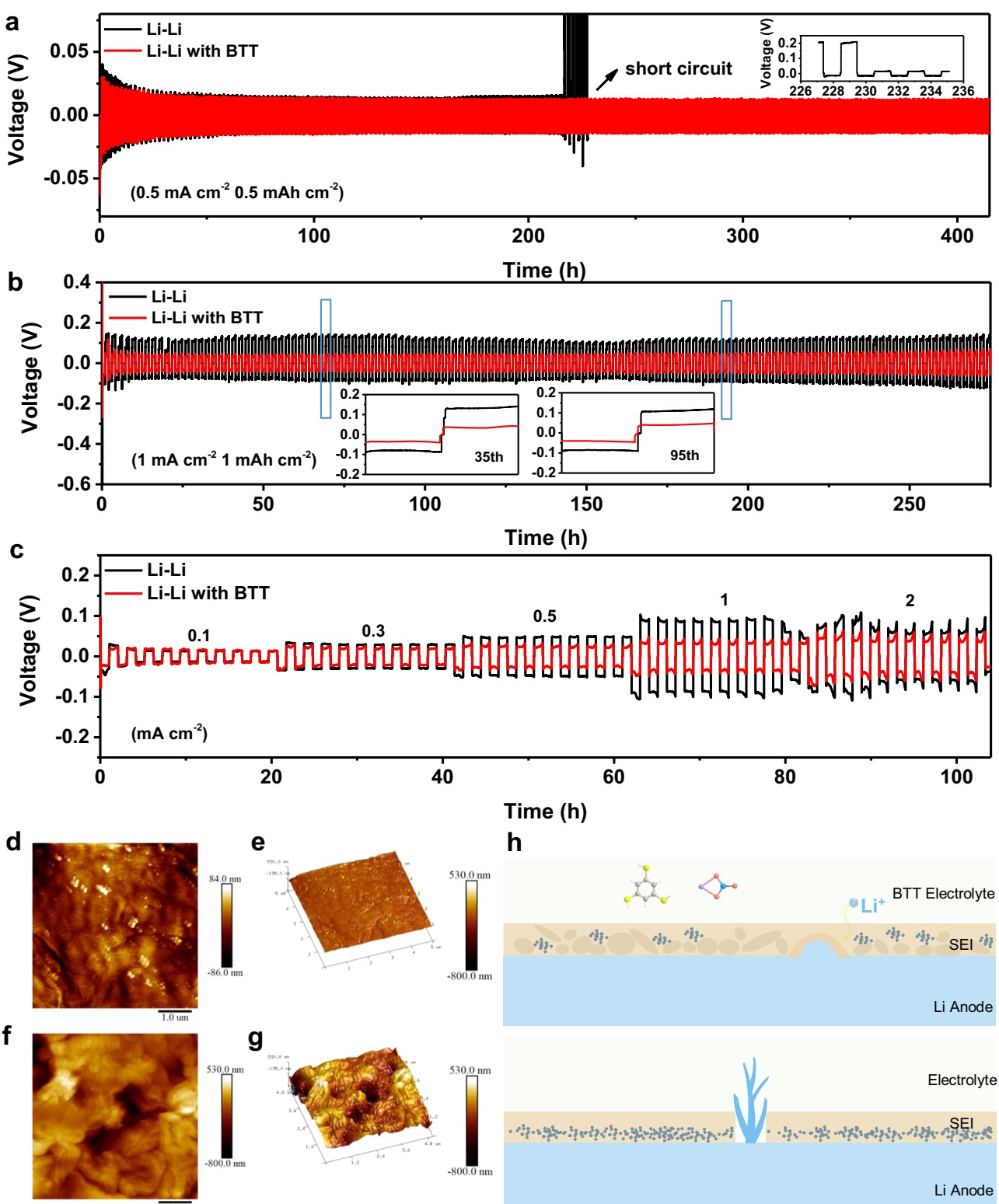

**Fig. 2 Long-term cycling performance of the symmetric Li–Li cells with BTT and blank electrolyte. a** Plating and stripping curves for the two types of symmetric cells with a current density of 0.5 mA cm$^{-2}$ and a capacity density of 0.5 mAh cm$^{-2}$. **b** Both types of cells at a current density of 1 mA cm$^{-2}$ and a capacity density of 1 mAh cm$^{-2}$. **c** Both types of cells at various currents and the cells cycled under 1 h charge and discharge conditions. **d, e** AFM images for the lithium foil with BTT electrolyte after 100 cycles with a current density of 0.5 mA cm$^{-2}$ and a capacity density of 0.5 mAh cm$^{-2}$. **f, g** AFM images for the control cell. **h** Schematic diagram of the flexible organic-inorganic SEI in BTT electrolyte and dendrite lithium in blank electrolyte.

morphologically, as shown in the SEM images (Supplementary Fig. 11a). The lithium metal in the BTT cell after 100 cycles at 0.5 mA cm$^{-2}$ (0.5 mAh cm$^{-2}$) shows a smooth and uniform plating layer with no cracks. In contrast, the one in the control cell shows a porous and incompact structure, indicating its continuous collapse during the Li plating/stripping process (Supplementary Fig. 11b). In addition, the SEI layer of the former (9.2 μm) is thinner and denser than that of the latter one (15.3 μm) (Supplementary Fig. 11c, d). Apart from this, SEM for the lithium foil in the BTT cell after 50 cycles (Supplementary Fig. 11e) also shows a more uniform deposition SEI surface than that in the control cell (Supplementary Fig. 11f).

Atomic force microscopy (AFM) analysis was used to investigate the mechanical properties of the SEI layer on the lithium metal electrodes in symmetric Li cell. As shown in Fig. 2d, e, the BTT electrode has a smooth, uniform, and non-dendritic lithium deposition after 100 cycles at 0.5 mA cm$^{-2}$ (0.5 mAh cm$^{-2}$). In contrast, the control electrode shows large granular features and clear cracks on the surface (Fig. 2f, g). The corresponding schematic diagrams of the deposition layers in the lithium are shown in Fig. 2h. The BTT electrode provides a smooth and uniform height/topography image, which indicates the continuous protection of the SEI layer during the Li plating/stripping process. The mechanical properties of SEI layers at plating stage were studied through indentation testing. Young's modulus can be obtained from the experimental indentation curves as shown in Supplementary Fig. 12, which was obtained by subtracting the cantilever defection from the total piezo drive distance. The BTT electrode has a higher Young's Modulus (2.255 GPa) (Supplementary Fig. 12a) than the control electrode in the commercial electrolyte (Supplementary Fig. 12b), indicating that the SEI formed with BTT retains better rigidity[42]. The rigid BTT SEI layer provides the ability to suppress the lithium dendrite growth and to cope with the change of volume stress in the deep charge/discharge state. Notably, the SEM and AFM results are consistent.

Chemical characterization of the two SEIs in symmetric Li cell after 10 cycles at 0.5 mA cm$^{-2}$ (0.5 mAh cm$^{-2}$) was conducted with X-ray photoelectron spectroscopy (XPS) depth profiling. The etching depth profile provides the chemical environment between Li and the S-H functional group of BTT based on the fitted Li, S, and F spectra. Figure 3a shows S 2p, F 1s, and Li 1s profiles acquired from the BTT electrode. Peaks in the Li 1s spectra can be attributed to Li$_3$-BTT (55.1 eV in Li 1s spectrum), Li–O (56.4 eV in Li 1s spectrum), and Li–F (57.6 eV in Li 1s spectrum)[43–45]. Both Li–O and Li–F come from the decomposition of LiTFSI, which is consistent with the control electrode. Li 1s spectra reveal that the peak of Li$_3$-BTT is always constant throughout the depth profile. Additionally, the relative concentration of Li–O and Li–F is reduced compared to the control electrode in the depth range of 58.8 nm to 117.6 nm, as shown in Fig. 3b. The appearance of the Li peak starting at a depth of 117.6 nm indicates that the SEI layer formed in the commercial electrolyte has been etched away and the deposited Li electrode underneath is detected. Peaks in the F 1s spectra are attributed to C–F and Li–F, giving the other half of the evidence. The signals of Li–F are reduced in the range between 58.8 nm and 117.6 nm, as shown in Fig. 3a, b, which is consistent with Li 1s spectra. Meanwhile, the Li$_3$-BTT spectral features are seen throughout the interface, and peaks attributed to S 2p and Li 1s appear together without any weakness. These results indicate that BTT was constantly reduced by Li at the interface during etching[38,39]. The atomic ratios of different elements at various sputtering depths for the BTT-SEI indicate a uniform layer with high sulfur content and uniform lithium distribution (Supplementary Fig. 13a and

species ratio in Supplementary Fig. 14a–c). The homogeneity of Li$_3$-BTT and the clear change of Li-F demonstrate that the BTT-SEI is composed of organic lithium and inorganic lithium compounds.

Combined with equation (1), a path for the interface layer formation can be described: the chemical reaction between BTT electrolyte and lithium foil comes from S-H bonds, and begins with their direct contact. Because of the trace gas from the reaction, irregular voids are left in the BTT–Li interface layer. During the process of charge/discharge, LiF particles created from the electrolyte decomposition fill in the voids of the BTT-Li interface layer gradually, forming the uniform and dense organic/inorganic interface layer (Supplementary Fig. 15)[46,47]. Hence, the BTT–Li interface layer provides more lithophilic sites for lithium plating to guarantee lithium transfer. Compared with the BTT electrode, the SEI formed in the commercial electrolyte is fragile and has poor ionic conductivity, with uneven composition (atomic ratio and species ratio are shown in Supplementary Figs. 13b, 14d, e). It also includes more reduced products, such as SO$_3$$^{2-}$ and S$^{2-}$ (from LiTFSI) in the S 2p spectrum[48,49], side reaction products of LiOH (55.0 eV in Li 1s spectrum), Li–O, and Li–F in the Li 1s spectrum (detailed binding energies are shown in Supplementary Table 4)[50,51]. Therefore, the presence of Li$_3$-BTT inhibits the adverse reactions and increases the Li$^+$ conductivity. Importantly, BTT can dynamically and instantaneously react with the exposed lithium surface to repair the damaged protective layer[52]. As evidenced from the experimental data, uniform lithium deposition, fast lithium-ion conduction, and integrated layer are achieved even at a high current[53,54].

**Properties of sulfur cathode with BTT-original interfacial layers.** To further obtain insight into the electrochemical transformation of the cathode side during cycling, further characterization methods were performed. 20 μL 0.15 M BTT electrolyte was added into the S cathode, some white materials appeared before the cell packaging (inset image in Fig. 4a). It indicates that BTT has reacted with the sulfur and the H$_2$S gas escapes subsequently. Fourier-transform infrared (FTIR) spectrum was tested to detect the changes of S–H bond. As shown in Fig. 4a, BTT exhibits a strong peak at 2550 cm$^{-1}$ that is attributed to the stretching vibration of the S–H group[39,55]. After the sulfur cathode was mixed with BTT, the S–H peak disappears. This proves that the S–H bond had undergone a chemical reaction. To validate the reaction of S–H bond, Raman spectrum was also collected from the mixed cathode and BTT (Supplementary Fig. 16a). The peak at 185 cm$^{-1}$ corresponds to the vibration of the S-H of BTT. In the mixed cathode, this S-H peak also disappeared, while new peaks are identified as the S-S bonds typical of a phenyl ring appear at 400–500 cm$^{-1}$[39].

To verify that the hydrogen of the S–H bond reacts with sulfur, leading to formation of a cathode electrolyte interphase layer between electrolyte and sulfur, the charged and discharged products were characterized. From the Raman spectra, the recharged BTT cathode shows the vibration of S–S bond at 460 cm$^{-1}$ and the C–S bond at 985 cm$^{-1}$[56,57], indicating that a new charged product appears (Fig. 4b). For the discharged BTT cathode, the peak of the S–S bond disappears and the peak of the C–S bond persists (due to the framework of the phenyl ring). The discharged product Li$_2$S has a low intensity Raman signal and can hardly be detected in the solution mode (Supplementary Fig. 16b)[57,58]. The disappearance of the characteristic peak of the S–H bond indicates the abstraction of hydrogen. Conversely, the appearance of the characteristic peak of the S–S bond comes from the recombination of the sulfur-sulfur bond between the

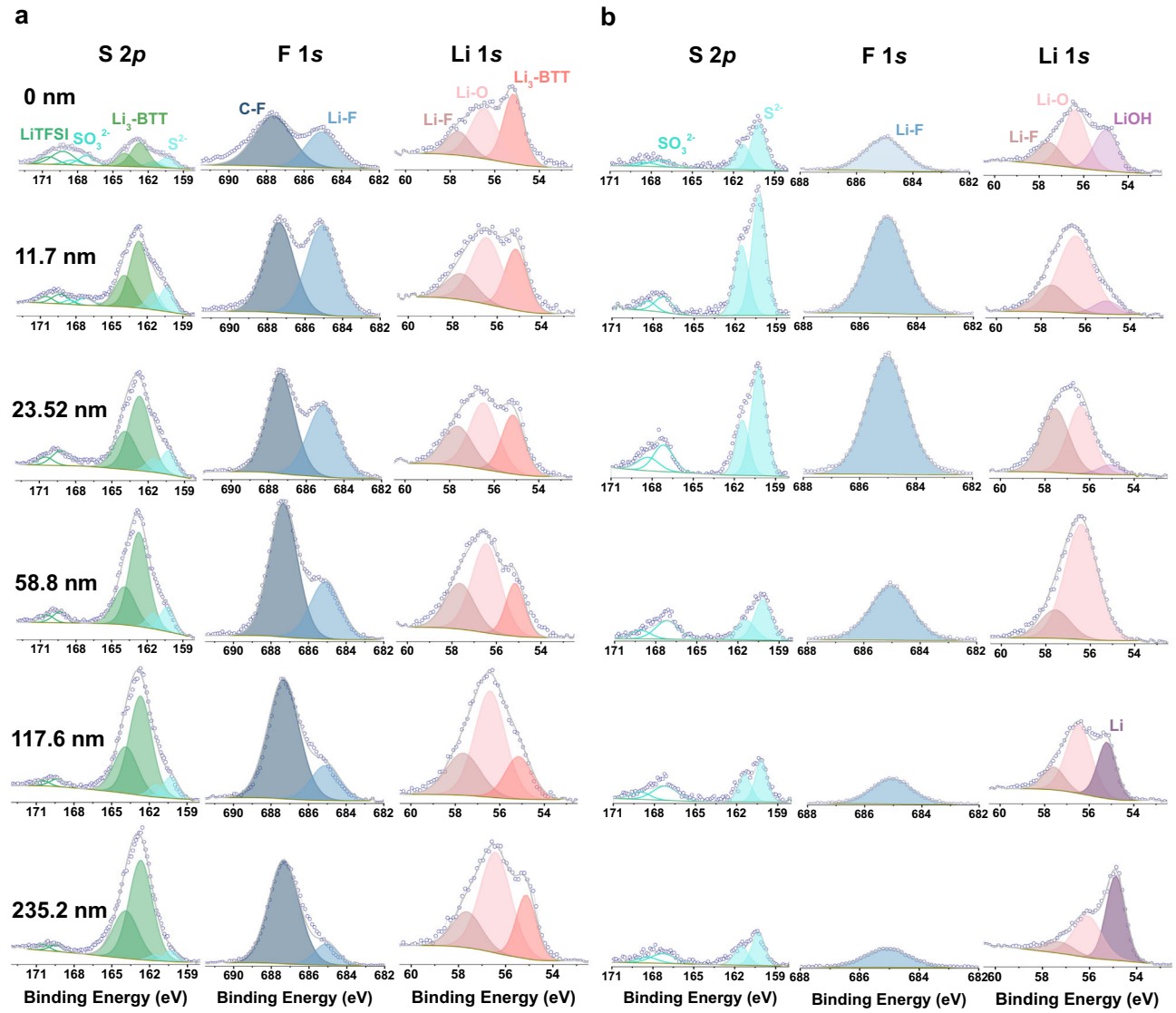

**Fig. 3 XPS spectra with different etching depths layer for lithium foils.** Two types of lithium foils were disassembled from symmetric cells with BTT and blank electrolyte after 10 cycles with a current density of 0.5 mA cm$^{-2}$ and a capacity density of 0.5 mAh cm$^{-2}$. **a** S 2p, F 1s, Li 1s spectra for the BTT symmetric cell, and **b** for the control cell.

phenyl ring (or $S_8$). From the FTIR spectroscopy (Supplementary Fig. 17), the peaks at 1,500 cm$^{-1}$ and 1,440 cm$^{-1}$ corresponding to the vibration of C–C bond in the phenyl ring almost disappear after recharged, due to the existence of bridged sulfur-sulfur bonds that limit the vibration of C–C bond for the phenyl ring. This demonstrates that the in situ oligomerization reaction between BTT and sulfur occurs.

In terms of the state of the electrode, the cathode material was sticky and adhered to the carbon nanotubes when the BTT cell was disassembled. No such phenomenon was observed in the control cell (Supplementary Fig. 18a, b). Transmission electron microscopy (TEM) obtained from the charged BTT cathode shows that a large amount of bulk material is attached to the carbon nanotubes of the BTT cell (Fig. 4c and Supplementary Fig. 19a), while the blank carbon nanotubes are smooth (inset in Fig. 4c). SEM also indicates that the charged products are wrapped and distributed in the network of carbon nanotubes (Supplementary Fig. 19b). The morphological changes are accompanied by the oligomerization during the charging procedure.

The electrochemical transformation of BTT-S cathode was explored by high-performance liquid chromatography coupled with quadrupole time-of-flight mass spectrometry (HPLC-QTof–MS) and XPS at the charged/discharged states. Figure 4d, e present the extract ion chromatogram (XIC) and the corresponding mass spectrum (MS) of the recharged product. The retention times of the main peaks are at 1.009 min and 1.058 min, corresponding to the mass/charge ratios of 377.8768 ($S_2PhS_3PhS_2$) and 441.8315 ($S_2PhS_5PhS_2$), respectively. XPS was collected to further ascertain chemical composition of the charged product (Fig. 4f). The S $2p_{3/2}$ peaks of the bridged S (S–S) and the S bonded to the phenyl ring (C-S) are found at 164.1 eV and 163.5 eV in the recharged product, respectively[49,55,56]. The presence of $Li_2S$ and $Li_2S_2$ are due to the incomplete charging of the cell. For the discharged product, Fig. 4g shows the retention time at 1.059 min corresponding to BTT (m/z 173.9638), which is due to the Li$^+$ in lithium benzenetrithiolate ($Li_3$-BTT) replaced by protons during the HPLC–QTof–MS measurement. Furthermore, the incompletely discharged intermediate product is shown in Supplementary Fig. 20a, b, which has a retention time at 1.746

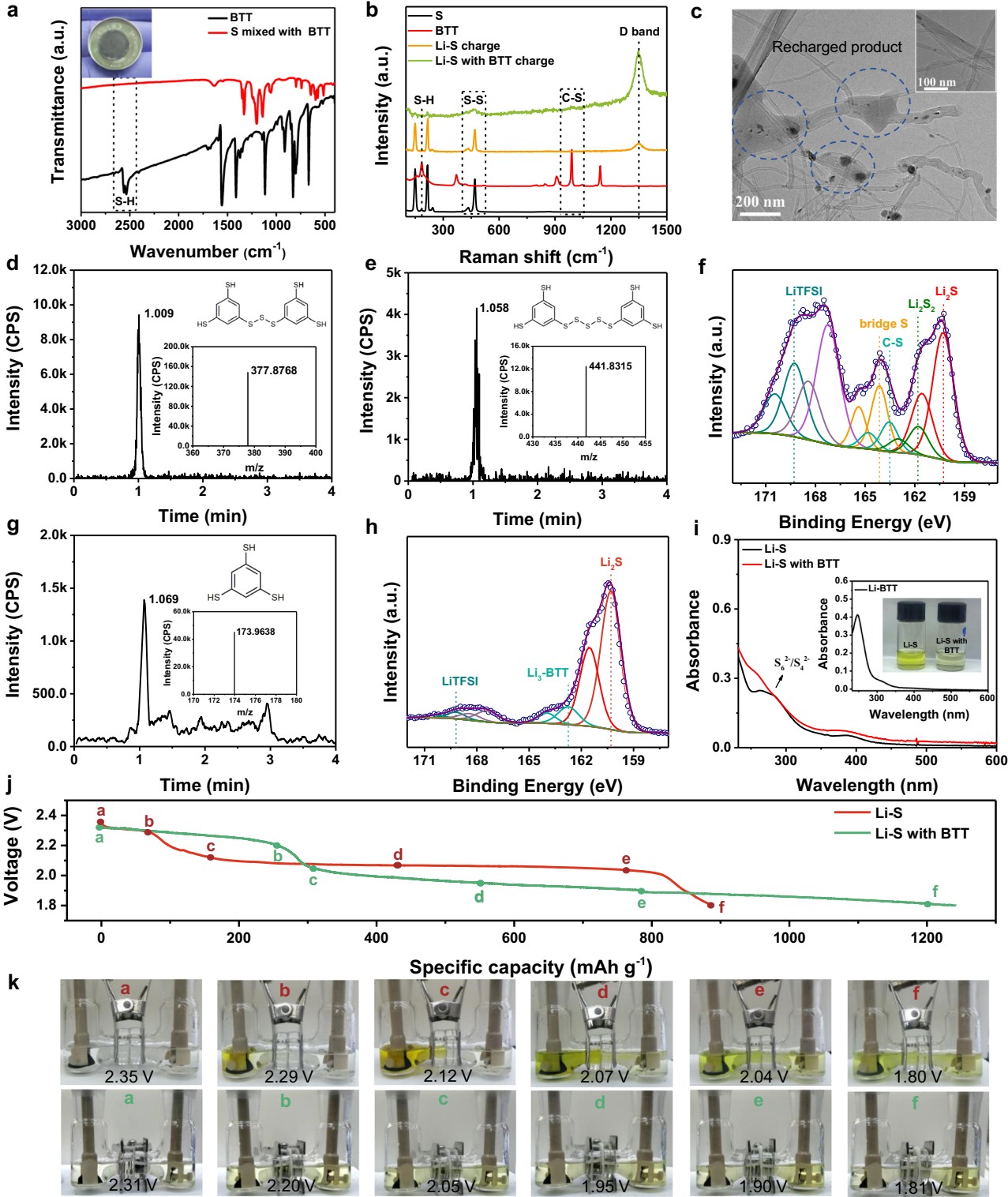

**Fig. 4 Characterization for the redox process in the cell. a** FTIR spectra of raw material BTT and sulfur cathode with 20 μL of 0.15 M BTT electrolyte. **b** Raman spectra of raw material BTT, elemental S, and the recharged products of the Li–S batteries with and without BTT. **c** TEM image of recharged cathode for Li–S cell with BTT electrolyte (inset image is the blank carbon paper). **d**, **e** XIC of the recharged products for BTT cell, (inset: the corresponding MS and structure). **f** XPS data for the recharged cathode after 5 cycles at 0.05C rate for the BTT cell. **g** XIC of the discharged product for BTT cell (inset: the corresponding MS and structure). **h** XPS spectrum of the discharged cathode after 5 cycles for BTT cell. **i** UV–vis absorption spectra of the solutions acquired by the Li–S cells with BTT and blank electrolyte discharge to 2.1 V. **j** First discharge curves for Li–S visible cells with BTT and blank electrolyte. **k** Photographs for the two types of cells in **j** at different discharge voltages.

min corresponding to the mass/charge ratio of 287.9120 ($PhS_6Li_3$). XPS reveals that the S $2p_{3/2}$ of discharged cathode for $Li_2S$ and $Li_3$-BTT are detected at 160.2 eV and 162.7 eV, respectively (Fig. 4h)[59–61]. Specific values of the binding energies are shown in Supplementary Table 4.

To further verify the improvement of the redox process, UV–Vis spectroscopy was conducted to detect polysulfides in the electrolyte. The two cells were cycled at 0.2C for several cycles and discharged to 2.1 V which the second discharge plateau for the $S_4^{2-}/S_6^{2-}$ transfer into $S_2^{2-}/S^{2-}$ in the Li–S cell appears, then the cathodes were sealed in the DOL/DME (1:1 v/v) solution for the test (image inset of Fig. 4i). As shown in Fig. 4i, the control cell has a broad peak at about 275 nm corresponding to the $S_4^{2-}/S_6^{2-}$ species[62–65]. The BTT cell has a peak at 250 nm which is due to the phenyl ring in the BTT. It is proved by the blank cell without sulfur (Li-BTT) (inset of Fig. 4i). This result indicates that BTT acts on sulfur and changes the redox path during charge/discharge. To further elucidate the discharge product of BTT ($Li_3$-BTT) effect on $Li_2S_x$, the $Li_3$-BTT is added into the $Li_2S_4$ solution in DOL/DME (1:1 v/v). The color of the $Li_3$-BTT solution is light yellow, but the color of the blank solution is dark red for the solvation of $Li_2S_4$ species (Supplementary Fig. 21a). The corresponding UV–Vis spectra of the solutions (Supplementary Fig. 21b) indicate that the $S_4^{2-}$ peak at about 275 nm disappears in the $Li_3$-BTT solution. Besides the UV–Vis results, the color changes of the two electrolytes are observed (Fig. 4k) at corresponding voltages (Fig. 4j) during the discharge process. Obviously, as the discharge progresses, the color of the blank electrolyte changes with the formation of the different-chain-length lithium polysulfides. The color changes from transparent to dark brown, then to green and finally to light yellow[66]. The color of the BTT electrolyte changes very little during the discharge process. Only at the beginning of the discharge, the electrolyte changes to be straw-colored, and then the color is not noticeably changed with the degree of discharge. It indicates the inhibition of lithium polysulfides and further confirms the new redox pathways by the introduction of BTT. In addition, $Li_3$-BTT also can react with polysulfides to promote their transformation to $Li_2S$. This demonstrates that BTT in the Li–S cell can mitigate the shuttle effect and improve the reaction kinetics of the cell.

From the analysis of the recharged and discharged products and the characterization of the redox process, the redox reactions of the BTT cell are formulated, as elaborated in Fig. 5a, b. During the 1st cycle, H from the S-H group is abstracted irreversibly. In the 1st recharge stage, the radical $PhS_3•$ and $S•$ are combined individually or in pairs to be oligomerized, then form new products **1** and **2**, as shown in the schematic (wavy line represents the S in benzene ring combine with S or H indeterminately). The first plateau results from a process where the $Li^+$ and $e^-$ attract the middle sulfur atom in the charged compounds $S_2PhS_3PhS_2$ **1** or $S_2PhS_5PhS_2$ **2**, then yield the intermediate **3** and **4**, respectively[67]. The lithiated intermediate products **3** and **4** are unstable and could react with $Li^+$ and $e^-$ quickly, corresponding to the short second plateau. The final step is the intermediate product ($PhS_6Li_3$) **5** is combined with 6 $Li^+$ and 6 $e^-$, forming the final products $Li_2S$ and lithium benzentrithiolate ($Li_3$-BTT) **6**. Therefore, the in situ oligomerization of BTT and sulfur changes the path of sulfur redox, forms the cathode electrolyte interphase, and prevents parasitic reactions. Density functional theory (DFT) calculations were also performed to better understand the possible reaction trajectory of the radical $PhS_3•$. After hydrogen abstraction from the S-H group, relatively strong attraction leads to an interaction between $PhS_3•$ and the Li atom. The Mulliken charge distribution of BTT before and after adsorption of Li are presented in Fig. 5c, which shows that more charge is transferred from Li to the S•. The ionic nature of the adsorbed Li atoms enables stronger

ionic bonding. After the charge transfer between the radical containing electronegative group and Li occurs, the interaction between them is further enhanced. It also suggests a general method: doping electronegative groups/atoms in the substrate improves the adsorption. DFT calculations also yield the electron density maps of both the highest occupied molecular orbitals (HOMO) and lowest unoccupied molecular orbitals (LUMO) for BTT, Li-BTT, $Li_2$-BTT, and $Li_3$-BTT. It is clear to see the HOMO energies of the lithiated products gradually increase from −6.72 to −6.57 eV from Li-BTT to $Li_3$-BTT species, and the LUMO energies present a down-hill character from −0.31 to −0.46 eV, showing a priority of both oxidation and reduction reactions.

## Discussion

In this work, we demonstrate that D-SEIs generated by the simple in situ interfacial reactions in the electrolyte containing BTT provide stable interfaces for the Li–S battery. The schematic illustration for the Li-S battery with BTT electrolyte is shown in Fig. 6. The D-SEIs are formed by the facile electrochemical/chemical reactions of the active S–H group. The SEI film composed of S–Li suppresses Li dendrite growth, improves the $Li^+$ conductivity, and possess a self-healing ability like a skin on the Li anode. The SEI film composed of S–S bonds causes oligomerization, changes the redox path of sulfur, and prevents the sulfur shuttle effect. The Li–S battery protected by the BTT additive demonstrates a high discharge capacity and a stable cycling performance with a high Coulombic efficiency and a superior rate capability. These results indicate that the major challenges encountered in the development of Li–S batteries can be effectively overcome. This research is prominent since it provides a simple strategy originated from bond chemistry, which fundamentally changes the Li plating/stripping behavior and redox path of sulfur cathode. It constructs an ingenious interface reaction to solve the inherent problems of batteries from a more chemical perspective. Given the existence of numerous available bonds reacting with Li or sulfur, it may yield more practical solutions to solve the intrinsic problems of Li–S battery through this strategy combined with computational approaches.

## Methods

**Materials**. Conventional Li–S electrolyte composed of 1 M LiTFSI and 0.15 M LiNO₃ in DOL and DME (1:1 v/v) was purchased from DoDoChem as the blank electrolyte. 1,3,5-benzentrithiol (BTT) was purchased from TCI (Shanghai) Development Co., Ltd and added to the conventional electrolyte to form the BTT electrolyte. Sulfur powder purchased from Aladdin was dissolved in carbon disulfide super dry solvent ($CS_2$) which was purchased from J&K, and 20 μL of the solution was injected into the Buckypaper (purchased from NanoTechLabs) and heated at 60 °C for 12 h to remove the solvent. The mass loading of sulfur was about 1.0 mg or 2–3 mg in the carbon paper. The carbon paper was punched into discs which 1.13 cm² about 2.0 mg (d = 12 mm), then dried in a vacuum oven for one day at 100 °C before use. Celgard 2400 as the separator was purchased from Celgard. All materials were purchased and used without further treatment.

**Electrochemical measurements**. CR2032 coin cells were fabricated in a (an argon (Ar)-filled) inert-gas glove box. For the half-cells, 20 μL BTT electrolyte was added to the cathode and then Celgard 2400 as a separator was placed on top of the cathode, then another 20 μL BTT electrolyte and lithium anode was added on the separator. For the control, 20 μL blank electrolyte (commercial Li-S electrolyte) was used instead on the two sides of the separator. To explore the capacity provided from BTT, the Li-BTT cell was assembled without the sulfur active material, only using 40 μL 0.15 M BTT electrolyte. The cells were crimped and taken out of the glove box and tested by LAND battery cycler between 1.8 and 3 V at different C rates (1C = 1675 mA g⁻¹). For the symmetric cell, CR2032 coin cell was fabricated with lithium foil on the positive case, then 15 μL 0.15 M BTT electrolyte was added on the two sides of the separator, then another lithium foil was placed on top, and the cell was crimped. For the control, 15 μL blank electrolyte was used for the lithium symmetric cell. Then the cells were taken out and cycled on the LAND battery cycler with different current density and capacity density. A Li–S pouch cell without electrolyte was provided from Zhong Ke Pai Si Energy Storage Technological Limited Company. The sulfur electrodes are double-coated on aluminum foil and the total sulfur mass in the cell is 2.8 g. The lithium metal anode is 100 μm

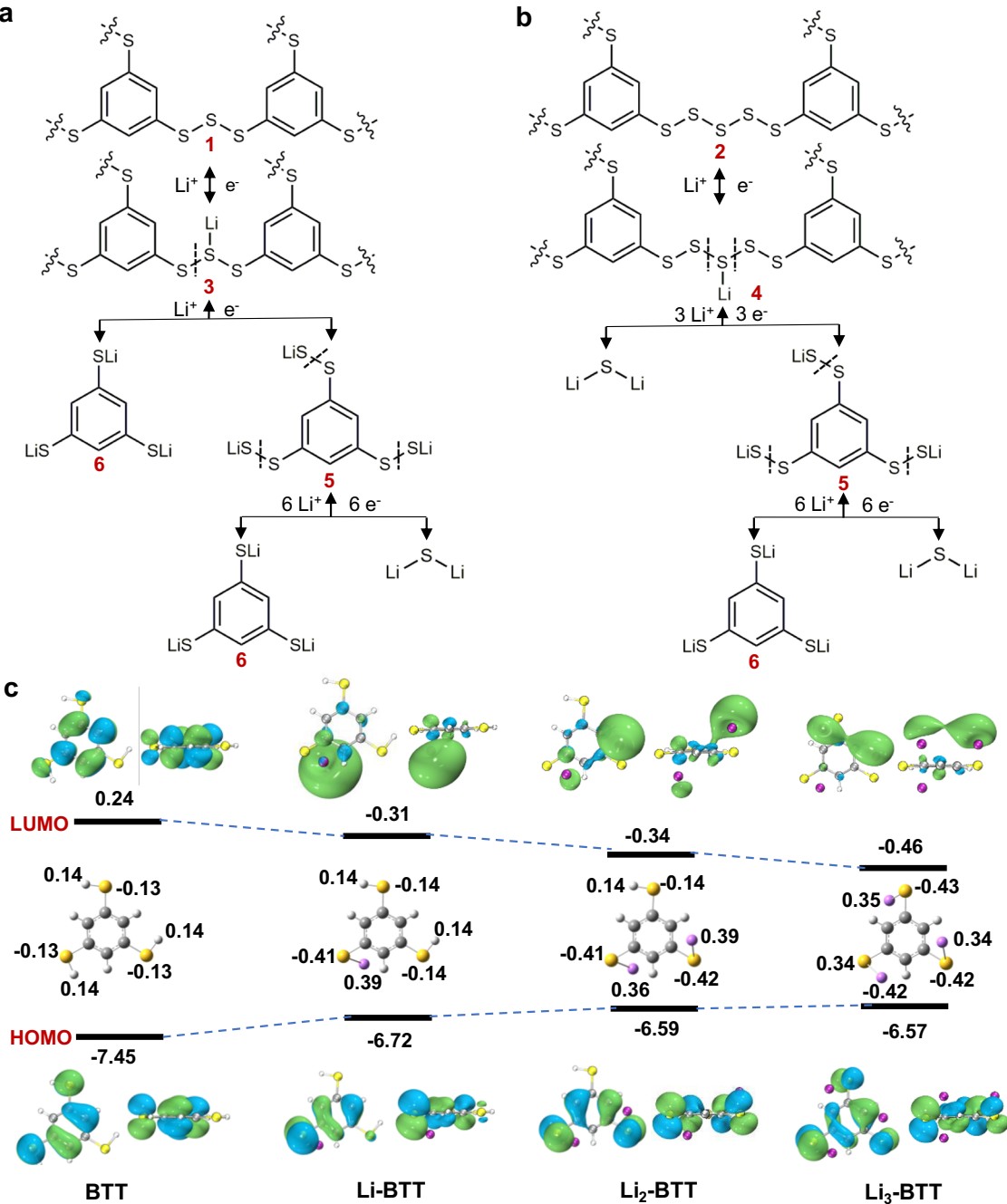

**Fig. 5 Scheme of redox process and molecular orbital energies. a, b** Redox reaction process of the Li-S rechargeable battery with BTT electrolyte. **c** The frontier molecular orbitals (MO) energies density and the Mulliken atomic charge distribution of state of the BTT and the lithiation products of BTT.

thick lithium foil. The pouch cell consists of 10 pieces of sulfur electrodes and 11 pieces of lithium metal foil. The BTT electrolyte was injected in the pouch cell and the cell was sealed in a dry room. After resting for 15 min, the generated $H_2/H_2S$ gases were discharged from the gas bag, then the pouch cell was sealed again. The pouch cell was rested for 24 h before testing at the current of 300 mA (0.1C rate). For the visible cell, 6 mL blank or the BTT electrolyte was used for the control or BTT cell with 1.3 mg S cathode, they were cycled at 0.1 mA to 1.7 V on the LAND battery cycler. Cyclic voltammetry (CV) and electrochemical impedance spectroscopy (EIS) were tested by the BioLogic VMP-3. For the EIS, the data was collected by the BioLogic VMP-3 with the frequency range of 1000 kHz–0.1 Hz. For the CV, the cell was swept from open circuit voltage (OCV) to 1.8 V and then back to 3 V at 0.05 mV s$^{-1}$. CVs was measured at different scanning of 0.03 mV s$^{-1}$, 0.05 mV s$^{-1}$, 0.07 mV s$^{-1}$, 0.1 mV s$^{-1}$, and 0.2 mV s$^{-1}$. The lithium-ion diffusion coefficient $D_{Li}^+$ is calculated by the CVs according to the Randles-Sevcik equation: $I_P = 2.69 \times 10^5 n^{1.5} A D^{0.5} C_{Li} v^{0.5}$. In the equation $I_P$ (A) is the peak current in CV, n is the number of the electron in the cell, A (cm$^{-2}$) is the electrode area, D (cm$^2$ s$^{-1}$)

represent the lithium-ion diffusion coefficient, $C_{Li}$ (mol mL$^{-1}$) is the lithium-ion concentration in the electrolyte, and v is the scanning rate (V s$^{-1}$).

**Characterization.** Scanning electron microscopy (SEM) of the cathode after 10 cycles at 0.05C was conducted with a Carl Zeiss Sigma 500 field emission. The lithium anodes for the symmetric cells with each type of electrolyte were first cycled at 0.5 mA cm$^{-2}$ 0.5 mAh cm$^{-2}$ for 100 cycles to deposit and strip lithium, then disassembled with the lithium foils removed and rinsed with dried DME solution prior to SEM. The lithium anode for Li–S cells with and without BTT were cycled at 0.5C for 50 cycles, then the lithium foils were rinsed with dried DME for test. X-ray photoelectron spectroscopy (XPS) of the charged and discharged cathodes after 5 cycles at 0.05C was performed with a 5000 VersaProbe 5000 VersaProbe II PS spectrometer with monochromatic Al Kα radiation. The XPS depth profile for the SEI layers formed in each type of electrolyte was conducted for the two symmetric cells after 10 cycles at 0.5 mA cm$^{-2}$ 0.5 mAh cm$^{-2}$, and the calculated depth

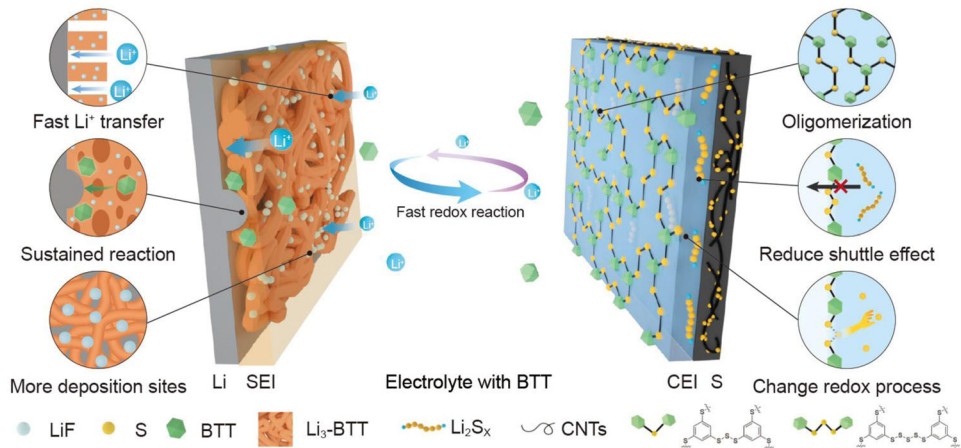

**Fig. 6 Schematic illustration for the Li-S battery with BTT electrolyte.** D-SEIs are formed on the interfaces of anode and cathode.

corresponds to the standard thermal oxidation of $SiO_2$ samples with the etching speed 5.88 nm $min^{-1}$. All the XPS samples were transferred by the vacuum device.

Atomic force microscopy (AFM) was performed for the symmetric cells of lithium foils with BTT and blank electrolyte to see the deposition condition. The cells were cycled at 0.5 mA $cm^{-2}$ 0.5 mAh $cm^{-2}$ for 100 cycles and then taken apart for examination using the Bruker DIMENSION ICON with ScanAsyst Atomic Force Microscope (AFM).

Transmission electron microscopy (TEM) was performed by the Tecnai G2 F20 microscope. The BTT cell was cycled at 0.05C and recharged for 5 cycles, then disassembled and the cathode carbon paper was placed into neat ethanol. Next, the cathode in the solution underwent ultrasonic dispersion and the resulting solution was placed on a copper net. The blank carbon paper was prepared with the same method. $^7Li$ nuclear magnetic resonance (NMR) was performed for the lithium bezenetrithiolate ($Li_3$-BTT) by the JNM-ECA600 system. First, 50 mg BTT was added to 1 mL DME, and then reacted with excessive lithium for 5 days in a sealed vial. Finally, the vial was opened to dry the solution and acquire the $Li_3$-BTT as a white powder, which was dissolved in the THF solution for the NMR test.

Fourier Transform Infrared (FTIR) transmittance was conducted by the NEXUS 470 infrared spectrometer with a scanning range is 400–4000 $cm^{-1}$, and the raw materials, sulfur cathode with 0.15 M BTT electrolyte, and the cycled cathode were ground with dried KBr powder and pressed into transparent discs. Raman samples of cells for each type of electrolyte were cycled at 0.05C after 10 cycles and then disassembled in the glove box. The cathode was taken for the test. The charged and discharged cathodes and the raw materials were examined by the HORIBA HR Evolution within a range of 100–1500 $cm^{-1}$ with a 532 nm laser at 5% ND filter.

Liquid chromatography-mass spectroscopy (LC-MS) to verify the charge and discharge product were performed on a Waters ACQUITY UPLC I-Class PLUS liquid chromatogram coupled to a Waters Xevo G2-XS QTof mass spectrometer. The cell cycled at 0.05C for 5 cycles and was then disassembled in the glove box. First, the charge and discharge cathodes were immersed in 2 mL mixture solution of chromatographic methanol and dimethyl sulfoxide (4:1 v/v) to let the active material dissolve in the solution. Then two drops of the solution using a 100 µL pipette were added to an UPLC-MS vial with 1 mL additional chromatographic methanol. The same method was used for all the samples in the LC-MS measurements.

Ultraviolet and visible spectrophotometry (UV-Vis) was performed by Agilent 8453 spectrometer with a range between 200 $cm^{-1}$ to 800 $cm^{-1}$ to characterize the polysulfide species in the cell. The different type cells were cycled at 0.1C for 10 cycles and finally discharge to 2.1 V and disassembled in the glove box, then the cathode was put in a solution of DOL and DME (1:1 v/v) for 1 h and were qual concentration diluted with the mixture solution. The mixture DOL/DME solution was used as a baseline correction. The Li-BTT cathode was tested with the same condition. In addition, the 5 mmol $Li_2S_4$ solution and the qual concentration $Li_2S_4$ solution with 4 mg $Li_3$-BTT were also detected in UV–Vis.

**Calculations**. The geometry optimization of the BTT, Li-BTT, $Li_2$-BTT, and $Li_3$-BTT were performed at the M06-2X/cc-pVTZ level of theory as implemented in the Gaussian 16 software[68], and the local minima were verified by the vibrational frequency analyses. The solvent effects were mimicked by the solvation model based on density (SMD) approach with the static dielectric constant of DME was set to $\epsilon = 7.07$[69,70].

## Data availability
The data underlying Figs. 1–4, Supplementary Figs. 1–5, 7-10, 12-14, 16, 17, 20, and 21 are provided as a Source Data file. The other data support the findings of this study are available from the corresponding author upon request. Source data are provided with this paper.

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

## Acknowledgements

This work was supported by the National Natural Science Foundation of China (Grant Nos. U2004214, 21975225, and 51902293).

## Author contributions

W.G. and W.Z. contributed equally to this work. Y.F. supervised the research. W.G. contributed the idea and designed the experiments. W.Z. carried out the experiments and materials characterization. Y.S. performed the theoretical calculations. W. G., W. Z., Y. S., D. W., Y. F., and A. M. wrote the manuscript. All authors discussed the results and commented on the manuscript.

## Competing interests

The authors declare no competing interests.
