## [Peer Review File · Nature Communications]

Reviewer #2 (Remarks to the Author):

The authors applied the 1,3,5-benzenetrithiol (BTT) as an additive into the sulfur cathode and lithium metal anode, which provides an effective SEI layer for Li-S battery performance. Furthermore, they explain how the materials affects in each electrode via various techniques. However, there are lots of missing points of the results and no brilliant idea or extremely high performance and scientific meaning. Therefore, I recommend the reject of publishing this paper on Nature communications.

1. In introduction, the mention of organic materials is not plausible for this research. Therefore, the author should describe the role of BTT as an additive in more detail and the specific purpose and direction of the study.
2. In the reaction mechanism of BTT with sulfur and lithium, the gas evolution is occurred during reaction. As the authors know, the evolution of H₂ or H₂S is one of the factors of battery performance decay. So, the author must explain the justification for this phenomenon, and this must be supported by experimental results.
3. In figure 1a, there is a clear difference in the voltage profile, especially at around 2.33 V (steep slope, expressed by the authors). However, there are any clear description of this steep slope in the manuscript. The difference of the voltage profile plateau must be explained because it is the most basic electrochemical results. (in the difference of the CV data, also)
4. In EIS spectroscopy, even in before cycles, there are two semicircles in Nyquist plots in BTT cell. It means that non-electrochemical reaction (just chemical reaction) is already occurred on the surface of BTT electrode before cycling. The authors should mention more clearly of these results.
5. In rate capability, the recent papers report the rate performance at high C-rates of over 3 C-rate. Please show the higher rate capability performance.
6. In high loading test, the author should present the areal capacity (mA cm⁻²) in high loading performance. And the voltage profile should be showed in supporting information.
7. In figure 4i (UV-Vis test), the author should represent the picture of test solution. Also, for more accurate experiment, the solution beaker cell test is more suitable for UV-Vis test. Because there is little electrolyte in coin cell, the obtain of lithium polysulfide is very difficult and unreliable. Therefore, the authors have to be found other approaches for showing these results.
8. Please check the typos. ((ex) Mathodes → Methods)
9. In Methods, I wonder the phase of BTT. Is it liquid or solid? If it is liquid, the liquid volume affects the electrolyte amount. So, please the details of the amount of electrolyte and calculation of ES ratio. If it is solid, please show the solubility of BTT in bare electrolyte. (the picture of 0.15M BTT electrolyte is very important, because non-soluble precipitates do not allow this experiments)
10. Recent reports about lithium-sulfur batteries utilized the lower voltage cut-off over at least 1.7 V, except special test such as high loading or high rate. However, in this manuscript, the author used the cut-off voltage of 1.8 V. Is there any reason?
11. In supplementary figure 4, the Li-S with BTT exhibited higher performance than Li-S. However, the cycling of the Li-S with BTT is unstable. please explain this.
12. In the diffusion coefficient part, please show the details of calculation of randles-sevcik equation (the parameters have to be shown). Typo : randles-sevick → randles-sevcik.
13. There is too much information in the manuscript. It is hard to find the direction or objective of this research. Some data sets are just array of data. I recommend that the paper is divided of two and each one should exhibit more specific and scientific logic.
14. Finally, the comparison of BTT additive with other additive studies is needed. The table of comparison should be added in supplementary information.

Reviewer #3 (Remarks to the Author):

In this manuscript, the authors reported the use of BTT as a bifunctional electrolyte additive to construct dual interfaces to protect both the lithium metal anode and sulfur cathode. However, this kind of strategy is novel and the overall quality of this manuscript can satisfy the standard of Nature Communications. I recommend this manuscript to be accepted after some minor revisions. The specific comments are described below:

1. The title of this manuscript is not appropriate as it does not indicate its application in Li-S battery area. Especially for those readers who are not in the battery area, this title is elusive.
2. In this manuscript, the authors mainly analyzed the chemical composition and chemical properties of SEI layer on the lithium metal in symmetric Li cells. However, the actual environment is rather different in Li-S full cell system. How about the properties of SEI layer on lithium metal in Li-S full cells?
3. It is suggested to show the performances under high sulfur loading and low E/S ratio (lean electrolyte).
4. In Li-S system, the theoretical capacity in the first discharge plateau is about 418 mAh g⁻¹ (occupy 25% of the theoretical capacity of sulfur), however, in Figure 1a, the capacity in the first discharge plateau has exceeded 450 mAh g⁻¹. Please explain this phenomenon.
5. Some recent publications related to Li metal anodes and Li-S batteries are recommended to be considered to compare the performances achieved in this work. These include: (1) Nature Communications 11 (2020) 5429; (2) Nature Communications 9 (2018) 3870; (3) Advanced Energy Materials 8 (2018) 1702485; (4) Angew. Chem. Int. Ed. 59 (2020) 9134; (5) Angew. Chem. Int. Ed. 58 (2019) 11364.
6. Does the amount of BTT affect the performance of Li-S batteries? It is recommended to optimize its concentration.

RESPONSE TO REVIEWERS' COMMENTS

REVIEWER 1:

Comment: Reviewer #1 in his confidential comment to us require you to provide electrochemical (cycling) performance of their Li-S cells with much lower E/S ratio, e.g., E/S=3.

Answer to comment: We thank the reviewer for the valuable suggestion. We understand the importance of low E/S ratio in Li-S batteries. In our study, we use bucky paper made of carbon nanotubes and carbon nanofibers as current collector, which is highly porous and can hold the electrolyte allowing full utilization of active material. However, it is very challenging to reduce the E/S ratio to 3, otherwise the electrode becomes very dry. However, we have now tried to reduce the E/S ratio to 5, as shown in the supplementary Figure 8b, the cell sustains 20 cycles. We hope the supplementary data can satisfy the request.

REVIEWER 2:

The authors applied the 1,3,5-benzenetrithiol (BTT) as an additive into the sulfur cathode and lithium metal anode, which provides an effective SEI layer for Li-S battery performance. Furthermore, they explain how the materials affects in each electrode via various techniques. However, there are lots of missing points of the results and no brilliant idea or extremely high performance and scientific meaning. Therefore, I recommend the reject of publishing this paper on Nature communications.

Answer: We thank the reviewer for the valuable comments. We have now provided substantial revisions and added significant amount of data to make our claims clear and strong. It can be concluded that the BTT additive does improve the performance of sulfur cathode and lithium metal anode. The thiol transformation and intriguing S-S bond chemistry can alter the redox reaction pathways leading to improved performance, which is reported for the first time. We believe this work will inspire more interest in functional electrolyte additives to enable advanced Li-S batteries and advance the Li-S battery technology to a new frontier.

Comment 1: In introduction, the mention of organic materials is not plausible for this research. Therefore, the author should describe the role of BTT as an additive in more detail and the specific purpose and direction of the study.

Answer to comment 1: Organothiols consisting of organic molecules with SH-groups are well suited to fabricate structurally layers, which can react with lithium metal and elemental sulfur, thus they are also promising electrolyte additives. 1,3,5-benzenetrithiol (BTT), as a typical organothiol, can not only react with Li to form SEI on the Li metal anode but also self-assemble with sulfur on the sulfur cathode to form stable and adaptable monolayer over homogenous areas. We have added this on page 3 in the revised manuscript.

Comment 2: In the reaction mechanism of BTT with sulfur and lithium, the gas evolution is occurred during reaction. As the authors know, the evolution of H_2 or H_2S is one of the factors of battery performance decay. So, the author must explain the justification for this phenomenon, and this must be supported by experimental results.

Answer to comment 2: We are addressing this question from the following three aspects. 1) The amount of BTT added to the positive or negative side is only $4.5 \mu\text{mol}$, resulting in a very small amount of H_2 or H_2S . We did a magnification experiment, as shown in Supplementary Fig. 1. A large amount of BTT (90 times more than that in the normal cell) were mixed with sulfur and a large amount of BTT (95 times more than that in the normal battery) were reacted with a piece of lithium metal in the DOL/DME solvent. Only at this dosage, the obvious deflating behavior can be captured. 2) BTT reacts with sulfur or lithium within few seconds. As shown in Fig. 4a, when BTT electrolyte was dropped onto the cathode, producing a white solid instantaneously. It indicates that BTT reacted with sulfur and H_2S gas escaped subsequently. Similarly, BTT reacted quickly with the lithium anode. The gas was released before the cells were encapsulated. 3) As for whether the residual gas generated during the cycle will affect the battery performance after the battery packaging, we have conducted a control test for justification as shown in Figure 1 (below). In this control experiment, the cathode and anode were pretreated with BTT electrolyte respectively, providing sufficient time and temperature for them to fully react. Then the pretreated electrodes were packaged and tested with a blank electrolyte without BTT. The experimental data show that the pretreated electrodes don't introduce any gas into the battery, but the Li-S cell with BTT electrolyte exhibits comparable performance and lower overpotential than the control cell. In short, the H_2 or H_2S generated during the cell preparation is negligible to be remained in the cell, which does not affect the battery performance.

Fig. 1 a Cycling performance of Li-S cells with BTT electrolyte and pretreated electrodes with blank electrolyte at 0.2 C rate. **b** Fifth charge and discharge curves of the two cells shown in a.

Comment 3:

In figure 1a, there is a clear difference in the voltage profile, especially at around 2.33 V (steep slope, expressed by the authors). However, there are any clear description of this steep slope in the manuscript. The difference of the voltage profile plateau must be explained because it is the most basic electrochemical results. (in the difference of the CV data, also)

Answer to comment 3: The BTT cell exhibits three discharge voltage plateaus including a plateau at 2.4 V, a steep slope in 2.3 V~2.1 V, a raised small peak at 2.1 V, and a long plateau at 2.1 V. The difference between BTT cell and the control cell (Li-S cell) is in the steep slope area. The BTT cell has a smaller slope, a larger discharge capacity, and a small bulge at 2.1 V. As we know, the traditional discharge curve of a Li-S cell is classified as: S_8 to Li_2S_x ($4 \leq x \leq 8$) (from 2.4 V), Li_2S_x ($4 \leq x \leq 8$) to Li_2S_x ($2 \leq x < 4$) (from 2.30 V~2.1 V), and Li_2S is formed (2.1 V). In 2.30 V~2.1 V region of the BTT cell, in addition to the transition of Li_2S_x ($4 \leq x \leq 8$), the S-S bonds of the BTT oligomer formed in the recharged process break, and bond with lithium ions and electrons, turning into Li_3 -BTT and Li_2S . The formed Li_3 -BTT completely precipitates at 2.1 V as shown in Fig. 1a, which is overlapped with the traditional sulfur transition of Li_2S_x ($4 \leq x \leq 8$) to Li_2S_x ($2 \leq x < 4$), causing the small peak at 2.1 V. On the contrary, because some of the sulfur is converted to Li_2S in advance, the 2.1 V plateau is shorter than that of the control cell. CV of the BTT cell has higher response current than the control cell. There is a small reduction peak contributed to Li_3 -BTT at 2.1 V, which is consistent with the discharge curve. We have made changes on pages 6 and 7 in red in the revised manuscript to make them clear.

Comment 4: In EIS spectroscopy, even in before cycles, there are two semicircles in Nyquist plots in BTT cell. It means that non-electrochemical reaction (just chemical reaction) is already occurred on the surface of BTT electrode before cycling. The authors should mention more clearly of these results.

Answer to comment 4: As mentioned in the answer to comment 2, some BTT already reacted with lithium and sulfur in the cell assembling process through chemical reactions (Fig. 4a), the semicircle in the high-to-medium frequency region is attributed to the interfacial impedance of the formed interfaces before cycling. After cycling, the reaction product of BTT with lithium and sulfur and the decomposition product of electrolyte form the SEI layer together through electrochemical reactions. We have made changes on page 7 in red in the revised manuscript to make it clear.

Comment 5: In rate capability, the recent papers report the rate performance at high C-rates of over 3 C-rate. Please show the higher rate capability performance.

Answer to comment 5: The C-rate performance of the Li-S with and without BTT cells with 2 C-rate is shown in Fig. 1e, the BTT cell has shown a better and more stable performance than the control cell even at higher C-rate. To further increase the C-rate, the low discharge voltage plateau will below the cutoff voltage of 1.8 V, causing partial discharge and limited capacities. Therefore, we didn't not test higher C-rate performance. We have added this on page 9 in the revised manuscript.

Comment 6: In high loading test, the author should present the areal capacity ($mA\ cm^{-2}$) in high loading performance. And the voltage profile should be showed in supporting information.

Answer to comment 6: The areal capacities in $mAh\ cm^{-2}$ have now been added in Fig. 1g in the revised manuscript and the corresponding voltage profiles have been added in Supplementary Figure 8a. Additional discussion has been added on pages 9 and 10 in the revised manuscript.

Comment 7: *In figure 4i (UV-Vis test), the author should represent the picture of test solution. Also, for more accurate experiment, the solution beaker cell test is more suitable for UV-Vis test. Because there is little electrolyte in coin cell, the obtain of lithium polysulfide is very difficult and unreliable. Therefore, the authors have to be found other approaches for showing these results.*

Answer to comment 7: The pictures of test solutions have now been added in Fig. 4i. The solution from the Li-S cell with BTT has a much lighter color than that of the control cell due to the reduced formation of lithium polysulfides. The suppression of polysulfide shuttling with the BTT was also visualized in glass cells shown in Figs. 4j and k. Obvious differences in the colors of the two electrolytes during the discharge process are observed (Fig. 4k) at corresponding voltages in Fig. 4j during discharge. Upon discharge, the color of the blank electrolyte changes from clear to dark brown, then to green, and finally to light yellow due to the presence of lithium polysulfides with different sulfur chain lengths and concentrations. The color of the BTT electrolyte changes very little at early discharge steps (point a to d in Fig. 4k) and then to straw-colored liquid at the end of discharge (point e to f in Fig. 4k), indicating the absence of lithium polysulfides and further confirming the new reaction pathways by the introduction of BTT. We have made corresponding changes on page 19-20 in the revised manuscript.

Comment 8: *Please check the typos. ((ex) Mathodes → Methods)*

Answer to comment 8: We apologize for these mistakes. We have worked on both language and readability and now corrected all the mistakes and typos. The revised sections are highlighted in red in the revised manuscript.

Comment 9: *In Methods, I wonder the phase of BTT. Is it liquid or solid? If it is liquid, the liquid volume affects the electrolyte amount. So, please the details of the amount of electrolyte and calculation of ES ratio. If it is solid, please show the solubility of BTT in bare electrolyte. (the picture of 0.15M BTT electrolyte is very important, because non-soluble precipitates do not allow this experiments)*

Answer to comment 9: BTT is a white powder and it can dissolve in electrolyte instantly. The picture of BTT powder and the BTT electrolyte are shown in Supplementary Fig. 2a. We have made changes on page 6 in the revised manuscript.

Comment 10: *Recent reports about lithium-sulfur batteries utilized the lower voltage cut-off over at least 1.7 V, except special test such as high loading or high rate. However, in this manuscript, the author used the cut-off voltage of 1.8 V. Is there any reason?*

Answer to comment 10: Low cut-off voltage of 1.7 V could result in the reduction of LiNO_3 in the electrolyte in the initial cycles, causing side reactions. We set the cut-off voltage at 1.8 V to make sure no side reactions occur. For a Li-S cell with BTT electrolyte, it was discharged to 1.7 V and the voltage profile in the 3rd cycle is shown below. It can be seen that almost no capacity can be achieved after 1.8 V in the discharge, therefore the cut-off voltage of 1.8 V is proper.

Fig. 2 Charge and discharge voltage profile of the Li-S cell with BTT in the 3rd cycle with the discharge cut-off voltage of 1.7 V at 0.2C rate.

Comment 11: *In supplementary figure 4, the Li-S with BTT exhibited higher performance than Li-S. However, the cycling of the Li-S with BTT is unstable. please explain this.*

Answer to comment 11: For the Li-S cells with BTT, BTT not only produces SEI on the lithium metal anode to protect it, but also is involved in the redox reactions of sulfur. The reaction between inorganic-organic hybrid materials has more unstable factors than the electrochemical reaction of single inorganic materials. Therefore, the cycling of the Li-S cell with BTT is unstable. Especially, the discharge capacity decreases in the first 10 cycles, which is because stable SEIs have not been formed yet. This is the most regrettable aspect of the BTT cell, but the final performance is still better than that of the control Li-S cell.

Comment 12: *In the diffusion coefficient part, please show the details of calculation of Randles-Sevcik equation (the parameters have to be shown). Typo : randles-sevcik → randles-sevcik.*

Answer to comment 12: The details of calculation of Randles-Sevcik equation have now been added on page 25 in the revised manuscript. We have also corrected the typos in the entire revised manuscript.

Comment 13: *There is too much information in the manuscript. It is hard to find the direction or objective of this research. Some data sets are just array of data. I recommend that the paper is divided of two and each one should exhibit more specific and scientific logic.*

Answer to comment 13: The suggestions from the reviewer help us to sort out the context of the article again, delete some redundant descriptions, and add subtitles, which can more clearly point to the logical relationship of the article. All the changes are marked in red. We also created a flow chart that would illustrate our theme more clearly (shown below). Based on the original intention of making the work free of omissions, the work has gone through a long period of experiments, accumulated a large number of data to support our claims, and also experienced repeated modifications before it appeared in the state. Therefore, deleting any data makes us feel

incomplete. Anyway, we have made adjustments as best we could, hoping to better represent our work.

Comment 14: Finally, the comparison of BTT additive with other additive studies is needed. The table of comparison should be added in supplementary information.

Answer to comment 14: The comparison of BTT additive with other additive studies has now been provided in Supplementary Table 2 and Supplementary Fig. 6. We have added this on page 9 in the revised manuscript.

REVIEWER 3:

Comment 1: The title of this manuscript is not appropriate as it does not indicate its application in Li-S battery area. Especially for those readers who are not in the battery area, this title is elusive.

Answer to comment 1: We agree with the reviewer's comment. We have now changed the article title to "Artificial dual solid-electrolyte interfaces based on *in situ* organothiols transformation in Li-S battery". We thank for the reviewer's suggestion.

Comment 2: In this manuscript, the authors mainly analyzed the chemical composition and chemical properties of SEI layer on the lithium metal in symmetric Li cells. However, the actual environment is rather different in Li-S full cell system. How about the properties of SEI layer on lithium metal in Li-S full cells?

Answer to comment 2: The SEM images of Li anode in Li-S full cells with and without BTT are shown in Supplementary Figs. 11e-f. The additive has a similar effect to form an inorganic-

organic SEI layer on the lithium metal surface, and guide a uniform lithium deposition in Li-S battery. We have added this on page 13 in the revised manuscript.

Comment 3: *It is suggested to show the performances under high sulfur loading and low E/S ratio (lean electrolyte).*

Answer to comment 3: The lower E/S ratio performance of high loading with areal current density has now been shown in Fig. 1f, and the corresponding charge and discharge curves are shown in Supplementary Fig. 8a. Higher sulfur loading of BTT cell with about 8 mg under different E/S ratios are shown in Supplementary Fig. 8b. We have made changes for the high loading cells on page 9-10 in the revised manuscript.

Comment 4: *In Li-S system, the theoretical capacity in the first discharge plateau is about 418 mAh g⁻¹ (occupy 25% of the theoretical capacity of sulfur), however, in Figure 1a, the capacity in the first discharge plateau has exceeded 450 mAh g⁻¹. Please explain this phenomenon.*

Answer to comment 4: The BTT cell exhibits three discharge voltage plateaus including a plateau at 2.4 V, a steep slope in 2.3 V~2.1 V, a raised small peak at 2.1 V, and a long plateau at 2.1 V. Maybe, the sections before 2.1 V can be called the first discharge plateau. The difference between BTT cell and the control cell (Li-S cell) is in the steep slope area. BTT cell has a smaller slope, a larger discharge capacity, and a small bulge at 2.1 V. As we know, the traditional discharge curve of a Li-S cell is classified as: S₈ to Li₂S_x (4≤x≤8) (from 2.4 V), Li₂S_x (4≤x≤8) to Li₂S_x (2≤x<4) (from 2.30 V~2.1 V), and Li₂S is formed (2.1 V). In 2.30 V~2.1 V region of the BTT cell, in addition to transition of Li₂S_x (4≤x≤8), the S-S bonds of the BTT oligomer formed in recharged process break, and bind lithium ions and electrons, turning into Li₃-BTT and Li₂S. The formed Li₃-BTT completely precipitates at 2.1 V as shown in Fig. 1a, which is overlapped with the traditional sulfur transition of Li₂S_x (4≤x≤8) to Li₂S_x (2≤x<4), causing the small peak at 2.1 V. On the contrary, because some of the sulfur is converted to Li₂S in advance, the 2.1 V plateau is shorter than that of the control cell. Hence, the first plateau of the BTT cell is not only a simple reaction for S₈ to Li₂S₄. It also covers the reactions of recharged oligomer products with lithium. So, the capacity in the discharge plateau has exceeded common values. We have now made changes on page 6 in the revised manuscript.

Comment 5: *Some recent publications related to Li metal anodes and Li-S batteries are recommended to be considered to compare the performances achieved in this work. These include: (1) Nature Communications 11 (2020) 5429; (2) Nature Communications 9 (2018) 3870; (3) Advanced Energy Materials 8 (2018) 1702485; (4) Angew. Chem. Int. Ed. 59 (2020) 9134; (5) Angew. Chem. Int. Ed. 58 (2019) 11364.*

Answer to comment 5: We have now cited these new and excellent works about Li-S battery and lithium protection in the revised manuscript (refs. 7, 10, 11, 12, and 31). These works have greatly inspired our work.

Comment 6: *Does the amount of BTT affect the performance of Li-S batteries? It is recommended to optimize its concentration.*

Answer to comment 6: The cycling performance of the Li-S cell with different concentration BTT is shown in Supplementary Figs. 2b-d. The Li-S cell with 0.15 M BTT has the lowest overpotential and highest capacity retention after 100 cycles at 0.5C rate. Hence, 0.15 M BTT is the optimal concentration. We have made changes on page 6 in the revised manuscript.

Reviewer #1 (Remarks to the Author):

The concept of using dual solid-electrolyte interfaces (D-SEIs) on both electrodes should be helpful for Li-S battery development. But the specific approach in this paper may not be practical - one has to look at this at large scale real application level. The reviewer's main concerns are:

1. As Reviewer #2 pointed out, H₂/H₂S will be an issue in the end. In small cells like coin cells, you won't see this kind of issues. Think about making large cells in manufacturing plants and applications with large battery pack together. Would it possible to use lithiated form of your additives to avoid H₂/H₂S?
2. The authors reports E/S ratio, with lowest of 5 μ L/mg which is still much higher for practical application. With increasing S loading and decreasing E/S ratio, cell capacity decreases with cycles very rapidly (Fig. 1, Fig. S8). This indicates the additive might not work for deep cycling or the consumption of additives with cycling which is not sustainable. A deep understanding on why the capacity fades so rapidly with increase S loading is needed.
3. The reviewer suggests large cell testing (>1Ah) for any new materials under relevant conditions for Li-S batteries.

Reviewer #2 (Remarks to the Author):

In the revised version, the authors have well addressed the concerns raised by the reviewers. Therefore, I recommend the acceptance of this work.

Reviewer #3 (Remarks to the Author):

The authors have comprehensively revised the manuscript according to my comments. I am satisfied with the revised version of the manuscript. Therefore, I would like to recommend to accept this manuscript for publication in Nature Communications.

RESPONSE TO REVIEWERS' COMMENTS

REVIEWER #1:

Overall Comment: *The concept of using dual solid-electrolyte interfaces (D-SEIs) on both electrodes should be helpful for Li-S battery development. But the specific approach in this paper may not be practical - one has to look at this at large scale real application level.*

Answer to overall comment: We thank for the valuable comments. We have now applied the BTT additive in a Li-S pouch cell (3 Ah) and provided additional data in Figure 1g to prove that this approach is practical. With BTT in the electrolyte, the pouch cell can be cycled stably over 17 cycles even with the very low E/S ratio of 2.6 $\mu\text{L mg}^{-1}$. The specific energy of electrodes and electrolyte reaches 441 Wh kg^{-1} in the 5th cycle. Corresponding discussion has now been added on pages 9 and 10 in the revised manuscript.

Comment 1: *As Reviewer #2 pointed out, $\text{H}_2/\text{H}_2\text{S}$ will be an issue in the end. In small cells like coin cells, you won't see this kind of issues. Think about making large cells in manufacturing plants and applications with large battery pack together. Would it possible to use lithiated form of your additives to avoid $\text{H}_2/\text{H}_2\text{S}$?*

Answer to comment 1: It is a good question. We have addressed this issue in the pouch cell. The pouch cell is made with additional empty space which is called “gas bag” to hold the gas generated from the cell, as shown in Figure S8b in the revised supporting information. When the BTT electrolyte was added in the pouch cell, $\text{H}_2/\text{H}_2\text{S}$ gases were obviously generated after 10-15 minutes. The gases were released by cutting the gas bag and then the cell was sealed perfectly. No more gases were generated afterwards even during the cycling process.

If the lithiated form of BTT is used to avoid $\text{H}_2/\text{H}_2\text{S}$, it cannot react with lithium metal anode and sulfur cathode to *in situ* form D-SEIs, which is not the aim of our work.

Comment 2: *The authors reports E/S ratio, with lowest of 5 $\mu\text{L}/\text{mg}$ which is still much higher for practical application. With increasing S loading and decreasing E/S ratio, cell capacity decreases with cycles very rapidly (Fig. 1, Fig. S8). This indicates the additive might not work for deep cycling or the consumption of additives with cycling which is not sustainable. A deep understanding on why the capacity fades so rapidly with increase S loading is needed.*

Answer to comment 2: The reason for the mentioned result is due to the high porosity of the carbon paper as current collector used in coin cells, which needs excess liquid electrolyte to maintain reasonable performance. To prove our approach, we have applied the BTT electrolyte in a practical pouch cell. The E/S ratio is reduced to 2.6 $\mu\text{L mg}^{-1}$ and the specific capacity of sulfur is still 1022 mAh g^{-1} at 0.1C rate in the 5th cycle. The pouch cell can be cycled over 17 cycles and the specific energy of the electrodes and electrolyte reaches 441 Wh kg^{-1} in the 5th cycle. Corresponding discussion has now been added on page 10 in the revised manuscript.

Comment 3: *The reviewer suggests large cell testing (>1Ah) for any new materials under relevant conditions for Li-S batteries.*

Answer to comment 3: We have applied the BTT electrolyte in the large pouch cell as described in the answers to previous questions.

REVIEWER #2:

Overall Comment: In the revised version, the authors have well addressed the concerns raised by the reviewers. Therefore, I recommend the acceptance of this work.

Answer to overall comment: We thank the reviewer for the positive comment and support.

REVIEWER #3:

Overall Comment: The authors have comprehensively revised the manuscript according to my comments. I am satisfied with the revised version of the manuscript. Therefore, I would like to recommend to accept this manuscript for publication in Nature Communications.

Answer to overall comment: We thank the reviewer for the positive comment and support.